# Innovative population-based strategies for primary prevention of cardiovascular disease: A 2-year randomised control trial evaluating behavioral change led by community champions versus brief advice

Delphine Le Goff[1,2]*, Gabriel Perraud[1,2], Mallaury Léon[3], Paul Aujoulat[1,2], Morgane Guillou-Landreat[2], Emmanuel Nowak[3], Marie Barais[1,2], Jean-Yves Le Reste[1,2]

**1** Department of General Practice, University of Western Brittany, Brest, France, **2** ER 7479 SPURBO, University of Western Brittany, Brest, France, **3** Clinical Research and Innovation Directorate, University Hospital of Brest, Brest, France

* docteurdlegoff@gmail.com

## Abstract

Cardiovascular diseases (CVD) caused 17.9 million deaths worldwide in 2019. General CVD prevention should be developed whilst controlling health expenditure. The aim of the SPICES project was to assess the efficacy of a community support intervention for adults with intermediate CVD risk at 24 months, compared to repeated brief advice alone. A randomised, controlled study was conducted in a poor, medically underserved, rural area in France from November 2019 to November 2021. Community champions lead small group sessions. They were specifically trained in behavioural change techniques and CVD prevention. Participants chose their own small, feasible cardiovascular health goals. The primary outcome was the difference in the Non-laboratory Interheart risk score (INL) in intention-to-treat at 24 months. Quality of life was assessed using the WHOQOL-BREF, diet using the DASH-questionnaire, physical activity using the IPAQ-short. Changes in BMI, smoking level, and self-declared alcohol consumption were recorded as health goals in the intervention group.1309 participants were eligible. 536 people were analysed. At 24 months, the difference Intervention–Control = -0.12 INL (95% CI -0.80 to 1.04) was not significant (p = 0.76). 257 people remained in the study. Most participants chose weight-loss as their objective. Although this study was impacted by the Covid-19 pandemic, pertinent observations were made. Participants spontaneously chose to lose weight, which was not an effective goal. The study was neglected by participants which suggests this population felt little concerned about CVD-prevention. Other awareness strategies should be developed. Public policies should be developed as individuals currently fail to improve their health.

**Trial registration**: This trial is registered at clinicaltrials.gov (NCT0388606).

**Data Availability Statement:** All relevant data are within the paper and its Supporting information files.

**Funding:** This research was funded by EUROPEAN COMISSION as a part of a Horizon 2020 project grant for the SPICES project. Project ID: 733356, funded under: H2020-EU.3.1.6—Health care provision and integrated care. The funders had no role in study design, data collection and analysis, decision to publish, or preparation of the manuscript.

**Competing interests:** The authors have declared that no competing interests exist.

## Introduction

Cardiovascular diseases (CVDs) caused 17.9 million deaths, which accounts for 32% of all deaths worldwide in 2019 [1]. Although the benefits of statins or angiotensin-converting enzyme inhibitors, have been clearly established over the last 30 years [2], reducing CVD age-standardised mortality is unequally distributed. Whereas mortality has decreased by 42% in high income countries, it has remained stable in developing countries.

Making behavioural changes to modifiable CVD risk factors [3] such as smoking, physical activity, eating a Mediterranean diet [4–7] can prevent CVDs. Behavioural change programs are effective, cost-effective and complement medication and have been successfully implemented in medically and economically-deprived areas [7] with established community health workers [8].

These programs and expertise from low and middle-income countries inspired the Scaling-up Packages of Interventions for Cardiovascular disease prevention in Europe and Sub-Saharan Africa (SPICES) consortium for economically-deprived, medically underserved rural areas in Europe and Africa [9]. The SPICES project delegated preventive care and involved communities to improve CVD risk factors.

Among the various methods for delivering behavioural change advice [10], volunteer community champions are an intensive behavioural change support. Also, brief advice is considered the gold standard comparator, and has been shown to be effective for smoking cessation [5] and is currently recommended to prevent risky alcohol consumption [11] and to modify CVD risk factors for people at a high CVD risk.

Central West Brittany, France, has a higher number of deaths from CVD and the lowest number of GPs and specialised physicians per capita compared to the French mean [12]. This makes it an economically-deprived, medically underserved rural area. Therefore, the French SPICES intervention was designed to integrate both strengths and weaknesses of the French health system and the specificities of the Central West Brittany area [13] to support behavioural change and reduce CVD risk factors.

The primary objective of this study was to evaluate the efficacy of a comprehensive community support intervention plus brief advice, for adults at moderate cardiovascular risk, to make behavioural change that improved cardiovascular risk, at 24 months, compared to brief advice alone, on the Interheart risk score (INL).

## Methods

### Trial design

SPICES trial is a randomized clinical trial with an open design (registered at clinicaltrials.gov; NCT0388606). Participant recruitment and follow-up were conducted between November 2019 and December 2021 in Central Western Brittany, France. The study was reported following Consolidated Standards of Reporting Trials (CONSORT) and its specific extension for related Covid-issues, the CONSERVE-CONSORT extension [14].

### Institutional review board statement

The study was conducted in accordance with the Declaration of Helsinki, and approved by the Comité de Protection des Personnes Sud Est IV CPP, Lyon, France: 18.12.14.72452; ID- RCB: 2018-A03201-54.

## Participants

First phase screeners identified potential participants during local social and sporting events organised in the summer of 2019, in collaboration with the local preventative health service. Those aged 18 years, living and/or working in central Brittany, with intermediate cardiovascular disease risk, (INL score 9–15) and willing to participate in the second phase were enrolled.

Participants were excluded if their INL score <9 (low cardiovascular risk), or high cardiovascular risk according to the INL (score strictly over 15) at enrolment.

## Randomisation and organisation

The two-year study period was organised in two time periods. Recruitment was planned during the summer months then enrolment and intervention started during the winter months. Once enrolled, participants were randomised in a 1:1 ratio to either the intervention or control group using the Redcap© software on which study data was also stored [15]. Due to the nature of the intervention, blinding was impossible to implement for the participants and the community champions. A significant effort was made to maintain blinding for the research team. The research team contacted potential participants two months after enrolment, independent of the first phase screeners, to form groups of similar geographical location, for community champions to run their groups.

## Intervention

The SPICES intervention consisted of asking participants to set small, feasible cardiovascular health goals (micro-objectives) to improve their CVD risk. In addition, lay people called community champions provided a short, medium, and long-term support program. These specifically trained community champions provided ongoing support during regular, ninety-minute motivational group sessions, delivered in local community halls. During these sessions, the community champions used motivational interviewing techniques and provided support to help participants formulate, construct and exploit feedback on their personal objectives. Between sessions, participants were expected to make behavioural micro-changes to achieve their chosen micro-objectives. Groups were expected to consist of 10 to 15 participants and 13 meetings were planned during the study period. The first three months the groups met every 15 days (Day 0, D15, D30, D45, D60). Then, two meetings were scheduled a month a part (M3, M4) then during the long maintenance phase meetings were scheduled every 3 months until study end (M7, M10, M13, M16, M19, M21).

All study participants received brief advice from the researchers at each study visit and a study newsletter to keep in contact. Additionally, they were invited to participate in three health conferences, tailored to the specific needs the community champions identified among their groups.

## Community champions recruitment and training

The first group of community champions was identified and recruited between January and November 2019 using local government stakeholders. In total, 37 individuals were approached, 21 were trained and 18 champions actually led intervention group sessions.

Champions were trained in motivational interviewing using a transtheoretical model approach which was appraised and delivered by national experts in behavioural change and motivation. Champions and research team members trialled this training version for improvements. Additional community champions were then recruited using a snowball strategy. This second group attended the modified 2-day, face-to-face training. After the intervention started,

champions were followed-up individually and in groups. Champions performed group interventions in geographical areas close to the participants place of residence or work.

## Outcomes

The primary outcome was improved cardiovascular risk, measured by the difference between the INL score for the intervention and control group at 24 months [16]. This validated score was chosen as it was fully clinical.

As secondary outcomes, quality of life was evaluated using the WHOQOL-BREF [17] questionnaire, diet modification with the DASH questionnaire [18] physical activity with the IPAQ-short [19] at 24 months. BMI reduction was assessed with calibrated tools, smoking level was counted, and modification of self-declared alcohol consumption was evaluated according to expert consensus as no internationally validated tool was found.

The validity and reliability of the measurement tools have been presented in a previous publication [13].

## Data collection

Outcome and quality of life data were collected at 0, 6, 12 and 24 months in line with significant milestones identified in the literature. This included baseline demographic and clinical characteristics, INL results, the number of sessions each participant attended and alcohol consumption. Additionally, micro-objectives were collected for the intervention group at 4 months and classified following the INL.

## Sample size

An inter-group comparative analysis was planned at the end of the intervention on the primary outcome on an intent-to-treat basis. A difference in the mean INL of 15% between the intervention and control group was expected. Based on this, the number of subjects was calculated for a power of 80%, a risk alpha = 0.05 and an effect size (mean difference/standard deviation) of 0.20. The number of subjects required was 394 per group, including 20% of lost to follow-up.

## Statistical methods

The statistical analysis was constructed in intention-to-treat for all included and randomised participants with an intermediate INL score (from 9 to 15). The main analysis was a two-step hierarchical analysis. First the average INL was compared between the two groups. If a statistically significant result occurred, the six secondary criteria would have been compared using the Holm-Bonferroni correction to allow multiple comparisons carried out at this stage.

The mean improvements (value at M24 minus value at M0) were compared between the two groups using an adjusted linear ANCOVA regression model. The adjusted difference should be interpreted as the difference that would have been between patients who started with the same baseline value.

Attendance was defined at 75% or more of the group meetings. Micro-objectives set at 4 months after the intervention started were recorded independently. Missing data were identified via the electronic CRF, and data collectors were called back to complete the missing data. The remaining incomplete files were declared as lost to follow-up. Analyses were performed using SAS, version 9.4 (SAS Institute, Inc., Cary, NC, USA).

## CONSERVE-CONSORT extension

**Extenuating circumstances.** The lockdown periods due to the Covid-19 pandemic from March to May 2020 and November to December 2020 and April to May 2021 substantially impacted study activities. Furthermore, municipal elections occurred in June 2020, during which key political stakeholders changed office without transferring SPICES project to the new political leaders. This was overcome with mediation between local government officials and the research team.

**Important modifications.** The intervention protocol was adapted to accommodate the contingencies resulting from the Covid-19 pandemic and changes to the political leadership. Data collection at M6 was delayed to M8, and an additional intermediate evaluation was added at 18 months to re-motivate all study participants.

**Responsible parties.** Protocol modifications were discussed locally with community champions, local government stakeholders and the research team. Modifications were then debated within the international team to ensure they were implemented in accordance with the SPICES principles. Modifications were then presented to the national ethics committee.

# Results and discussion

## Participant flow diagram

Fig 1 illustrates the study flow diagram.

## Baseline characteristics

Overall, 1309 willing people were eligible for the study of which, 536 were randomised. Baseline characteristics are presented in Table 1. Extensive characteristics, including secondary outcomes, are presented in S1 Appendix. The mean age of the cohort was 59.8 ± 14 years, and most were female (64.6%). The mean INL was 11.85 ± 2.04, the mean BMI was 27.06 ± 5.01. No heavy drinkers were included.

## Results at 24 months

At 24 months, 110 and 147 people remained in the intervention and control groups (total = 257). The decrease in INL score was lower in the Intervention group (0.46) compared to the Control group (0.55), which was not significant (difference Intervention -Control-0.12 (-0.80–1.04) $p = 0.76$).

At M8, M12, M18, M24 no significant differences in the INL score were observed either (Table 2).

## Secondary outcomes

Exploratory analysis of WHOQOL-BREF, DASH questionnaire IPAQ-short questionnaire results did not reveal any significant difference between quality-of-life measures, alcohol consumption or BMI (S2 Appendix).

## Implementation outcomes

29% of eligible subjects agreed to participate of which, 48% completed the study. Participant attrition occurred throughout the study. Overall, 119 participants responded and set micro-objectives (Fig 2). Most participants chose a micro-objective to lose weight but 21% of participants also expressed goals that were external to INL parameters.

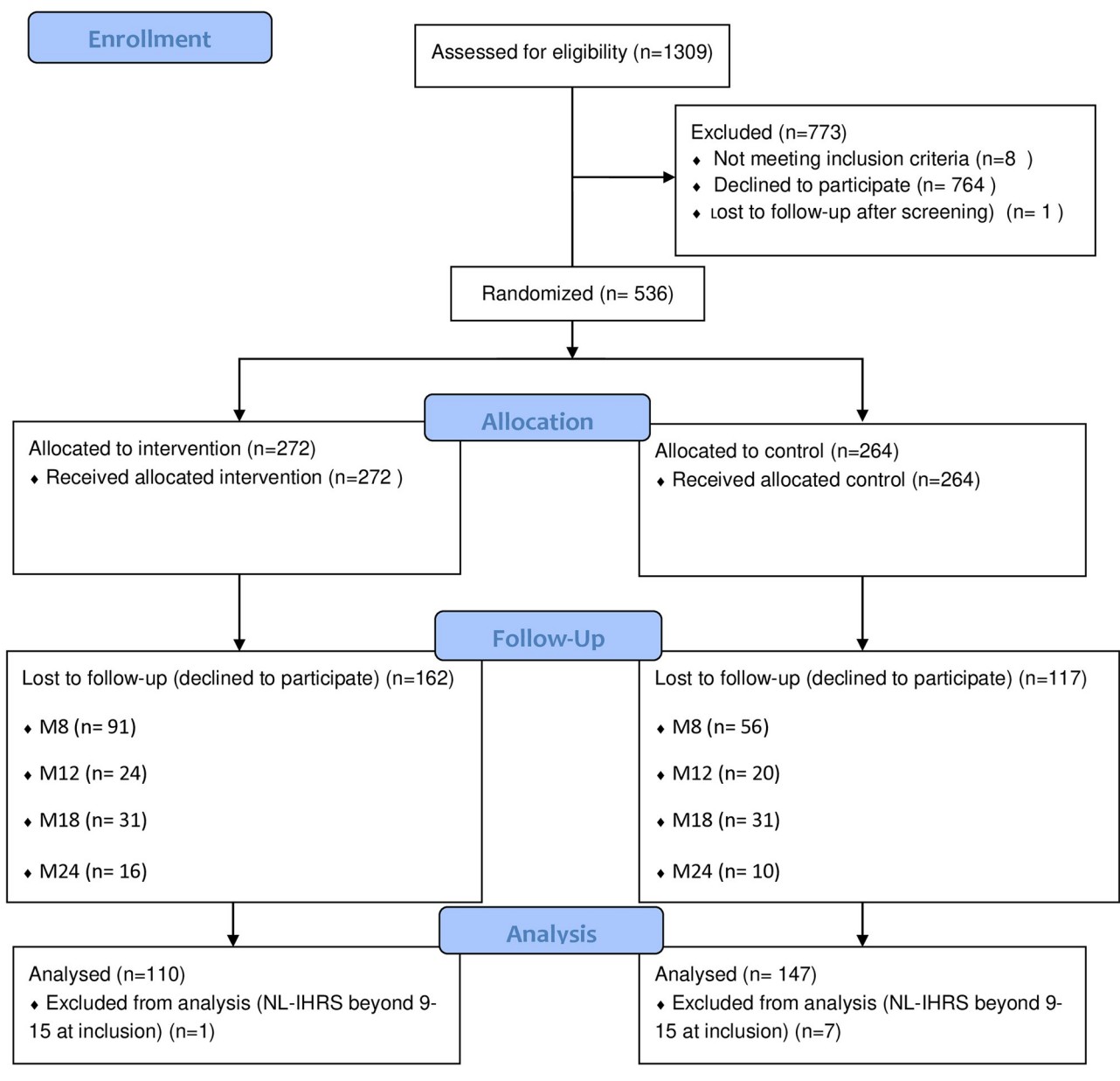

**Fig 1. CONSORT flow diagram of the SPICES participants.**

No participant reached the goal of thirteen meetings and the best group achieved nine sessions. Details of attended sessions are presented in Fig 3.

No harm or unintended effect occurred in either group.

## Main results

The SPICES behavioural change intervention did not result in a significant difference in INL-CVD risk measure between the Intervention and Control groups. Interestingly, most participants in the Intervention group set themselves a goal to lose weight and, participation was extremely low throughout the study compared to the expected estimates.

**Table 1. Baseline demographic, clinical characteristics of the SPICES participants and baseline INL.**

| Variable | | Total (N = 536) | Control (N = 264) | Intervention (N = 272) |
|---|---|---|---|---|
| Age | Mean ± (Min-Max) | 60 (46–74) | 59.0 (44–74) | 60 (47–73) |
| Gender | Male | 190 (35.4%) | 96 (36.4%) | 94 (34.6%) |
| BMI | Mean ± Standard deviation | 27.06 ± 5.01 | 26.70 ± 4.60 | 27.42 ± 5.35 |
| Over the last 30 days, how many days did you drink alcohol? | Median (Min-Max) | 4 (1–10) | 4 (1–10) | 4 (1–10) |
| On average, how many standard drinks did you drink per day? | Median (Min-Max) | 1 (0–2) | 1 (0–2) | 1 (0–2) |
| Over the last 30 days, how many days did you drink more than 6 alcohol standard drinks per day? | Median | 0 | 0 | 0 |
| **Interheart** | | | | |
| Smoking. Pick the description which matches you best: | I never smoked | 247 (46.1%) | 115 (43.6%) | 132 (48.5%) |
| | OR I am a current smoker, or I smoked regularly in the last 12 months, | 85 (15.9%) | 41 (15.6%) | 16 (5.9%) |
| | OR I am a former smoker (last smoked more than 12 months ago) | 204 (38.1%) | 108 (40.9%) | 96 (35.3%) |
| Second hand smoke Over the past 12 months, what has been your typical exposure to other people's tobacco smoke? | Less than 1 h or exposure per week or no exposure | 467 (87.1%) | 233 (88.3%) | 234 (86.0%) |
| | OR one or more hours of second-hand smoke exposure per week | 69 (12.9%) | 31 (11.7%) | 38 (14.0%) |
| Do you have diabetes mellitus? | Yes | 13 (2.4%) | 6 (2.3%) | 7 (2.6%) |
| Do you have high blood pressure? | Yes | 174 (32.5%) | 72 (27.3%) | 102 (37.5%) |
| Family history: Have either or both of your biological parents had a heart attack? | Yes | 140 (26.1%) | 72 (27.3%) | 68 (25.0%) |
| Psychosocial factors: How often have you felt work or home life stress in the last year? Pick one only | Several periods or permanent stress | 311 (58.0%) | 156 (59.1%) | 155 (57.0%) |
| During the past 12 months, was there ever a time when you felt sad, blue, or depressed for two weeks or more in a row? | Yes | 166 (31.0%) | 84 (31.8%) | 82 (30.1%) |
| Do you eat savoury food or snacks one or more times a day? | Yes | 15 (2.8%) | 7 (2.7%) | 8 (2.9%) |
| Do you eat deep fried foods or snacks or fast foods 3 or more times a week? | Yes | 28 (5.2%) | 20 (7.6%) | 8 (2.9%) |
| Do you eat fruit one or more times daily? | No | 128 (23.9%) | 75 (28.4%) | 53 (19.5%) |
| Do you eat vegetables one or more times daily? | No | 91 (17.0%) | 51 (19.3%) | 40 (14.7%) |
| Do you eat meat and/ or poultry 2 or more times daily? | Yes | 117 (21.8%) | 55 (20.8%) | 62 (22.8%) |
| How active are you during your leisure time? | I am mainly sedentary or perform mild exercise (requiring minimal effort) | 174 (32.5%) | 90 (34.1%) | 84 (30.9%) |
| Waist-to-hip ratio | Mean ± Standard deviation | 0.90 ± 0.09 | 0.90 ± 0.09 | 0.90 ± 0.09 |
| Interheart Score | Mean ± Standard deviation | 11.85 ± 2.04 | 11.79 ± 2.01 | 11.90 ± 2.07 |

This negative result may be attributed to extrinsic and intrinsic issues with this study. Firstly, the lockdowns imposed during the Covid-19 pandemic diminished sporting and social activities, preventing patients from achieving micro-objectives that involved physical activity. Furthermore, this is supported by contemporary observational research that identified an increased weight gain during this time [20, 21], mostly among women who declared more snacking [20–22]. Also, the lockdowns contributed to participant attrition, lowering the power of the study and contributing to the lack of effect observed. However, this disengagement may not be entirely due to the consequences of the COVID-19 pandemic. Interestingly, before pandemic began, the levels of engagement differed among the community champions. Some champions were unable to maintain regular contact with their small groups, even during the intensive phase, whereas other champions were able to organise nine meetings before the first lockdown and maintain an individual link with participants during subsequent lockdowns.

**Table 2. Analysis of the INL at M24.**

| Criteria | INTERVENTION Group | CONTROL Group | *Adjusted difference |
|---|---|---|---|
| | Mean (standard deviation) | Mean (standard deviation) | INTERVENTION–CONTROL |
| | (N = 272) | (N = 264) | Mean (95% CI) |
| M0 | 11.90 (2.07) | 11.79 (2.01) | |
| | N = 272 (0 missing) | N = 264 (0 missing) | |
| M6 | 11.18 (3.10) | 11.04 (3.06) | |
| | N = 181 (91 missing) | N = 208 (56 missing) | |
| Δ (M6-M0) | -0.66 (2.68) | -0.70 (2.94) | 0.07 (-0.48–0.62) |
| | | | p = 0.81 |
| M12 | 11.94 (3.80) | 11.72 (3.89) | |
| | N = 157 (115 missing) | N = 188 (76 missing) | |
| Δ (M12-M0) | 0.04 (3.44) | -0.18 (3.78) | 0.22 (-0.54–0.98) |
| | | | p = 0.57 |
| M18 | 11.51 (3.52) | 11.67 (3.91) | |
| | N = 126 (146 missing) | N = 157 (107 missing) | |
| Δ (M18-M0) | -0.40 (3.29) | -0.20 (3.90) | -0.18 (-1.02–0.65) |
| | | | p = 0.67 |
| M24 | 11.55 (3.84) | 11.35 (4.05) | |
| | N = 110 (162 missing) | N = 147 (117 missing) | |
| Δ (M24-M0) | -0.46 (3.68) | -0.55 (3.81) | 0.12 (-0.80–1.04) |
| | | | p = 0.80 |

This difference in engagement and in cohesive qualities was unexpected and highlights the need to identify appropriate qualities for a successful champion role. To understand this further, a qualitative study is currently ongoing to identify these appropriate specific qualities. Similarly, a French study published in 2019 tested educational sessions delivered by general

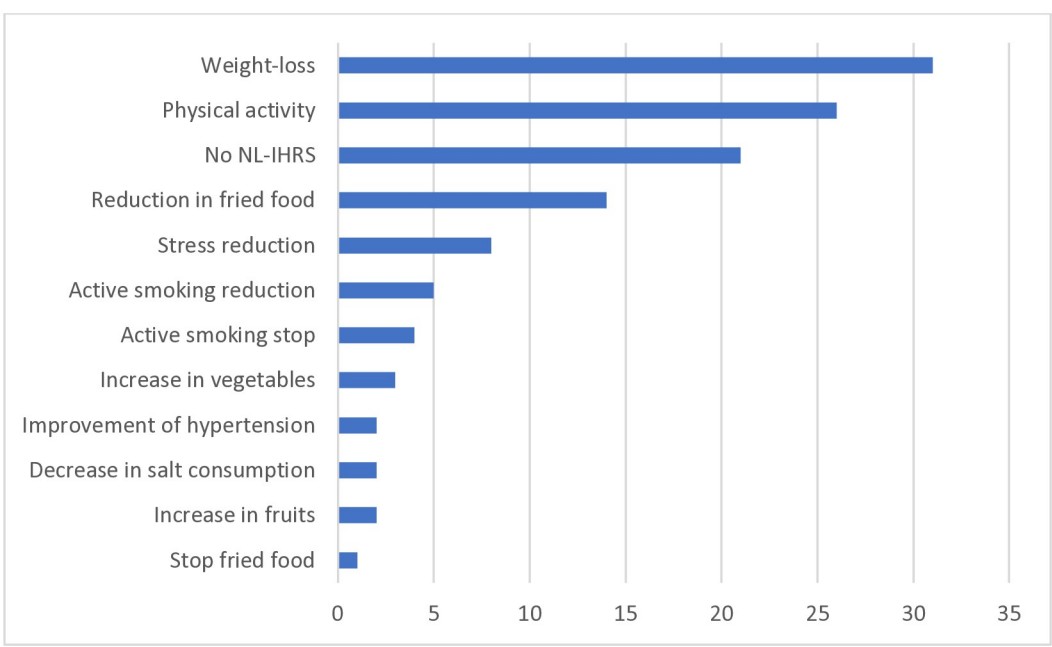

**Fig 2. Micro-objectives, according to the INL, participants chose in % at 4 months.**

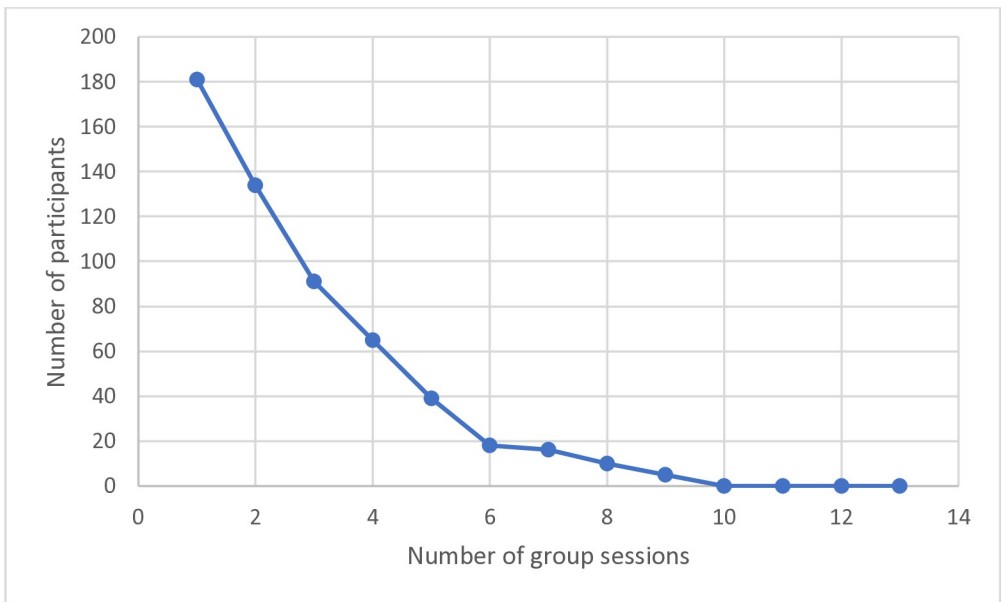

**Fig 3. Number of participants attending each session.**

practitioners, health educators, dieticians, and sport educators. No difference was found in cardiovascular risk between personalised or group-based therapeutic education [23]. The authors suggested the behavioural change model may be more responsible for the efficacy of the intervention than the person delivering the intervention. This still advocates for community champions involvement.

Possibly, broader measures that include community wide actions may be required for a behavioural change model to have an effect. It has been shown in a Swedish study that evaluated the effectiveness of a behavioural change intervention combining an individual strategy (invitations and cardiovascular health checks) with a community-wide strategy. These community-wide activities created a positive dynamic, using health messaging in the media and food retailers and introducing a food labelling system. This intervention effectively decreased the CVD risk [24].

Apart from issues involved with implementing the intervention, the appropriateness of using the INL as a primary measurement tool for cardiovascular risk is questionable. INL was chosen for its ease of use in a community setting, its having heavily weighted items like smoking and lighter weighted items like diet that participants could easily improve. Nevertheless, the INL did not capture small improvements in stress and depression. Similarly, another preventive study found that the waist-to-hip ratio, included in the INL to evaluate obesity as a cardiovascular risk, did not change [25].

Interestingly, most participants in the SPICES Intervention group set themselves a goal to lose weight. Yet the mean BMI was 27.06± 5.01, which is slightly higher than the mean BMI for women (24.2) and men (25.5) in France in 2014 [26]. Notably, weight-loss for people with a BMI of around 27 is not recommended, because they risk having weight rebound and greater weight gain [27]. Moreover, published studies on weight-loss strategies often show very small benefits with no change in obesity/overweight class and involve small populations [25, 28]. In comparison, 16% of the cohort were current smokers, and only 25% of Intervention group smokers tried to stop smoking. Yet quitting smoking is a change which provides major health

benefits whereas health benefits from losing weight are lower. This learning from the SPICES project is crucial when promoting health amongst the general population. There is a need to support people in their health choices to redirect their decisions towards effective strategies.

## Strengths

The two main strengths of SPICES are the general public population recruited in a local community setting, and the large sample size for a behavioural change study to address cardiovascular risk factors. Previously, research has been limited to in-patients, were outdated or had small populations [29], limiting their generalisability.

Also, measuring changes at 24 months, by which time a behavioural change would be expected to be stabilised, was a key benefit of the study. Typically, many behavioural studies have 6-month endpoints which may overestimate the effect of lifestyle changing Interventions. When behavioural improvement relaxes, the effect of interventions for weight-loss, physical activity or smoking cessation have been shown to wane [30].

## Limits

**Selection biases.** Although the screening targeted the general public in places and events where an equal distribution of men and women were expected, the cohort was mainly female (64.6%). This limits its generalisation. However, this female overrepresentation may not have affected the study results. Although more women were recruited, and women tend to be preoccupied by their health, behaviour did not change [31]. Also, the mean age of the cohort (60 years) was also older than the mean age in Central Brittany (43 years). This could be explained as people younger than 18 were excluded and study participation required participants to attend events during their free time, which was easier for retirees than young, active population.

Attrition bias is a major issue in this study due to a lower than the expected number of enrolments, a large number of unexpected dropouts early in the study. Despite the intervention being free of charge implemented out of working hours, and close to participant living area, 59% of eligible participants chose not to participate. When writing a research protocol involving behavior change, it is commonly accepted that an attrition of 20% should be anticipated. However, attrition rates in motivational studies are extremely variable. Many studies do not provide their attrition rate. A Chinese study aimed at improving blood pressure over 6 months reported an attrition rate of 0.08% [32]. At the extreme opposite, a Scandinavian 12-month study aimed at improving professional activity in the workplace had an attrition rate of 78% [33]. In the SPICES study, non-participants reported competing priorities of different kinds. They could be family caregivers, have new professional constraints, or have a non-cardiovascular health event. In qualitative interviews, some dropouts were related to the time-consuming nature of the intervention, the nature of group intervention, a feeling of déjà vu of cardiovascular prevention messages, the relationship with champions [34, 35]. Maybe participants with moderate cardiovascular disease were less motivated to make behavioural changes. Most previous studies on behavioural change selected participants with high cardiovascular risk who are most likely to notice an immediate, measurable health benefit.

**Confusion biases.** In France, the Covid- 19 related health measures included three strict lockdowns and gradual deconfinements, restricting public meetings for months. Group meetings were interrupted at each lockdown and resumptions were slowed by local stakeholders. The qualitative study of dropouts highlighted the difficulty of maintaining physical activity due to lockdowns, an upheaval of personal priorities induced by the pandemic and a saturation of prevention messages induced by the omnipresence of messages to limit the transmission of

Covid [34, 35]. Citizens and participants did not wish to dematerialize the meetings. Data collection was dematerialized, allowing collection to be ensured according to the protocol.

## Conclusion

If procedures inspired by the SPICES project should be performed again, participants could be stratified with personality questionnaires. Researchers should identify people capable of change. Moreover, stronger community involvement should be encouraged, with multiple levels of intervention: media, retailers, restaurants, schools. These interventions have proven to be effective and costless. Finally, strategies like the SPICES project are insufficient alone. The massive investment in training time and money to support participants in these programs did not lead to a better outcome, despite the abundant published sociological literature asserting the opposite.

This strongly suggests there is a need for public policies to defend the health interests of the population. Regulating salt and saturated fatty acids in industrial and artisanal foods would be invisible to the population but be effective whereas individual strategies have failed to improve population health.

The resource-intensive SPICES behavioural change program led by community champions was not effective in lowering INL-CVD risk compared to brief advice only. Investment in public regulations to improve food quality may be more effective than expecting individuals alone to improve their cardiovascular health outcomes.

## Supporting information

**S1 File.**
(DOCX)

**S1 Checklist. CONSORT 2010 checklist of information to include when reporting a randomised trial*.**
(DOC)

**S1 Appendix. CONSERVE checklists.**
(DOCX)

**S2 Appendix. Analysis of secondary outcomes.**
(DOCX)

## Acknowledgments

This article was supported by the French network of University Hospitals HUGO ("Hôpitaux Universitaires du Grand Ouest"). We deeply thank Mrs Jiaying Zhang, English native speaker, for the revision of this manuscript and Amy Whereat BSc UNSW for medical writing assistance. We also thank Elise Poulhazan for data management.

## Author Contributions

**Conceptualization:** Delphine Le Goff, Morgane Guillou-Landreat, Jean-Yves Le Reste.

**Data curation:** Delphine Le Goff, Gabriel Perraud, Jean-Yves Le Reste.

**Formal analysis:** Delphine Le Goff, Gabriel Perraud, Mallaury Léon, Emmanuel Nowak, Jean-Yves Le Reste.

**Funding acquisition:** Jean-Yves Le Reste.

**Investigation:** Delphine Le Goff, Gabriel Perraud, Jean-Yves Le Reste.

**Methodology:** Delphine Le Goff, Gabriel Perraud, Jean-Yves Le Reste.

**Project administration:** Delphine Le Goff.

**Supervision:** Jean-Yves Le Reste.

**Validation:** Delphine Le Goff, Gabriel Perraud, Jean-Yves Le Reste.

**Visualization:** Delphine Le Goff.

**Writing – original draft:** Delphine Le Goff, Gabriel Perraud, Marie Barais, Jean-Yves Le Reste.

**Writing – review & editing:** Delphine Le Goff, Gabriel Perraud, Mallaury Léon, Paul Aujoulat, Morgane Guillou-Landreat, Emmanuel Nowak, Marie Barais, Jean-Yves Le Reste.

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
