## [Decision Letter · Decision Letter 0]

26 Jul 2024

PONE-D-23-42871Innovative population-based strategies for primary prevention of cardiovascular disease: A 2-year randomised control trial evaluating behavioral change led by community champions versus brief advice.PLOS ONE

Dear Dr. Le Goff,

Thank you for submitting your manuscript to PLOS ONE. After careful consideration, we feel that it has merit but does not fully meet PLOS ONE’s publication criteria as it currently stands. Therefore, we invite you to submit a revised version of the manuscript that addresses the points raised during the review process.

We look forward to receiving your revised manuscript.

Kind regards,

Mohd Akbar Bhat

Academic Editor

PLOS ONE

Journal Requirements:

“This research was funded by EUROPEAN COMISSION as a part of a Horizon 2020 project grant for the SPICES project. Project ID: 733356, funded under: H2020-EU.3.1.6—Health care provision and integrated care.”

Please state what role the funders took in the study.  If the funders had no role, please state: 'The funders had no role in study design, data collection and analysis, decision to publish, or preparation of the manuscript.'

3. In this instance it seems there may be acceptable restrictions in place that prevent the public sharing of your minimal data. However, in line with our goal of ensuring long-term data availability to all interested researchers, PLOS’ Data Policy states that authors cannot be the sole named individuals responsible for ensuring data access (http://journals.plos.org/plosone/s/data-availability#loc-acceptable-data-sharing-methods).

Reviewers' comments:

Reviewer's Responses to Questions

**Comments to the Author**

1. Is the manuscript technically sound, and do the data support the conclusions?

Reviewer #1: Yes

Reviewer #2: Yes

Reviewer #3: Partly

2. Has the statistical analysis been performed appropriately and rigorously? 

Reviewer #1: Yes

Reviewer #2: Yes

Reviewer #3: I Don't Know

3. Have the authors made all data underlying the findings in their manuscript fully available?

Reviewer #1: Yes

Reviewer #2: Yes

Reviewer #3: Yes

4. Is the manuscript presented in an intelligible fashion and written in standard English?

Reviewer #1: Yes

Reviewer #2: Yes

Reviewer #3: Yes

5. Review Comments to the Author

Reviewer #1: Some Minor Revision

This randomised trial was obviously carried out in challenging circumstances created in part by COVID restrictions. As I can imagine the investigators will be disappointed by the outcome suggesting little difference between the Control and Targeted interventions. Nevertheless, their results are very important to present and, no doubt, many lessons learned.

Some suggestions for simplifying their presentation of the results are given below.

Page Lines

2 16 Not clear in the Abstract if −0.12 represents Intervention – Control or the reverse difference. (see also Page 8, Paragraph 2, line 3 which gives ‘+0.12’!)

16 Suggest that ‘the difference in INL was not significant (-0,12 (-0,80; 1,04) p=0,758)’ would be better presented ‘the difference Intervention – Control = -0.12 INL (95% CI -0.80 to 1.04) was not significant (p = 0.76).’

I have replaced the ‘virgule’ by ‘.’ to be consistent with your Table 2 and Appendix B.

16 Usual to quote p-values to two significant figures, so in the above ‘0.76’ rather than ‘0.758’. This also applies to the many p-values quoted in Table 2 & Appendix B.

6 10-13 Sentence starting ‘Then … ’ can be omitted as the Holm-Bonferroni corrections seem unnecessary here as all they are likely to do is increase the size of the (here non-significant) p-values.

14-5 This too can be omitted

6 15-16 The statistical methods section states ‘The mean improvements (value at M24 minus value at M0) were compared between the two groups using an adjusted linear ANCOVA regression model.’. However, I wonder if a simpler analysis would be easier to understand (see ** below)

8 §3.2, 3-4. This sentence implies that statistical significance tests were conducted on baseline differences between intervention groups. Since the two interventions were randomised, then any differences in Table 1 characteristics are therefore, by definition, random. So statistical testing is not relevant here

18 Table 1 In variables such as age, BMI and several others usually more informative for descriptive purposes to give the minimum and maximum values rather than quoting the SD.

For gender and many variables below the numbers are integers so the decimal points should be omitted.

Also for variables using the median, better to use the minimum and maximum values rather than the IQR.

** Table 2 I suspect the unadjusted differences in the final column of this table would be very similar to the adjusted values. Adjusted values might be helpful if they substantially change the Unadjusted values and hence substantially change the interpretation of the findings. I suspect ‘substantial changes’ will not be the case here.

I recommend, the authors to consider quoting the Unadjusted values in Table 2 (and in Appendix B).

If these changes are made then Page 6, Statistical Methods, lines 2-5 and 8-9 can be simplified.

Reviewer #2: Excellent idea, it trigger a new perspective, the research was conducted in an efficient way, results are clear enough, discussion section is written in a very good manner, conclusion is also clearly mentioned. Well done.

Reviewer #3: The manuscript is well done. Here are some concerns that should be addressed:

The significant drop-off in participant numbers from eligibility to analysis (1309 eligible vs. 536 analyzed) raises concerns about participant engagement and retention. Discussing the reasons for attrition and potential biases introduced by this attrition is crucial for interpreting the study's findings accurately.

The specific impacts of the COVID-19 pandemic on the study, such as disruptions to intervention delivery, participant follow-up, and data collection, should be elaborated upon. How these challenges were managed, and their potential influence on study outcomes need to be thoroughly discussed.

Despite community champions leading behavioral change sessions, the primary outcome (INL score) did not show a significant difference between intervention and control groups at 24 months. Potential reasons for this lack of significant effect, including the nature of the behavioral goals chosen by participants and the intensity or duration of the intervention, must be discussed.

Clarify the validity and reliability of the measurement tools used to assess outcomes, such as the WHOQOL-BREF for quality of life, the DASH questionnaire for diet, and the IPAQ-short for physical activity. Ensuring these tools are appropriate and validated for the study population strengthens the reliability of reported findings.

6. PLOS authors have the option to publish the peer review history of their article (what does this mean?). If published, this will include your full peer review and any attached files.

Reviewer #1: No

Reviewer #2: No

Reviewer #3: **Yes: **Mohammed Ahmed Akkaif

---

## [Author Response · Author response to Decision Letter 0]

11 Sep 2024

Response to Reviewers

Reviewer #1: Some Minor Revision

This randomised trial was obviously carried out in challenging circumstances created in part by COVID restrictions. As I can imagine the investigators will be disappointed by the outcome suggesting little difference between the Control and Targeted interventions. Nevertheless, their results are very important to present and, no doubt, many lessons learned.

Some suggestions for simplifying their presentation of the results are given below.

Page Lines 2 16 Not clear in the Abstract if −0.12 represents Intervention – Control or the reverse difference. (see also Page 8, Paragraph 2, line 3 which gives ‘+0.12’!)

16 Suggest that ‘the difference in INL was not significant (-0,12 (-0,80; 1,04) p=0,758)’ would be better presented ‘the difference Intervention – Control = -0.12 INL (95% CI -0.80 to 1.04) was not significant (p = 0.76).

We thank the reviewer for this rewording that we have adopted. The abstract has been modified accordingly as the 3.3 paragraph.

I have replaced the ‘virgule’ by ‘.’ to be consistent with your Table 2 and Appendix B.

We thank the reviewer for this replacement. We replaced the commas in the numerical values and checked the entire manuscript carefully.

16 Usual to quote p-values to two significant figures, so in the above ‘0.76’ rather than ‘0.758’. This also applies to the many p-values quoted in Table 2 & Appendix B.

Thanks to the reviewer for this comment. Every p-value has been expressed with two significant figures.

6 10-13 Sentence starting ‘Then … ’ can be omitted as the Holm-Bonferroni corrections seem unnecessary here as all they are likely to do is increase the size of the (here non-significant) p-values.

14-5 This too can be omitted

6 15-16 The statistical methods section states ‘The mean improvements (value at M24 minus value at M0) were compared between the two groups using an adjusted linear ANCOVA regression model.However, I wonder if a simpler analysis would be easier to understand (see ** below)

We thank the reviewer for his cautious review of the statistical section. We agree that ANCOVA is somewhat more complicated than a Student’s t-test. However the statistical methods have to be prespecified and must not be data driven. We planned (in our Statistical Analysis Plan) to perform the analyses of improvements with adjusment for baseline according to the European Medicines Agency recomandations (EMA, 2015, Guideline on adjustment for baseline covariates in clinical trials §5.6 ‘Change from baseline’ analyses). The use of change from baseline with adjustment for baseline is generally more precise than change of baseline without adjustment. If we had performed the analysis without adjustment, the poor choice of the analysis method could be criticized, even if it does not actually substantially change the interpretation of the findings.

Nevertherless, we have reduced the description of secondary statistics that we were unable to achieve. Sentence 10-13 has been reworded and sentence 14-15 has been deleted. In order to enlighten non-statistician readers, we added the following sentence : « The adjusted difference should be interpreted as the difference that would have been between patients who started with the same baeline value ».

8 §3.2, 3-4. This sentence implies that statistical significance tests were conducted on baseline differences between intervention groups. Since the two interventions were randomised, then any differences in Table 1 characteristics are therefore, by definition, random. So statistical testing is not relevant here. 

We agree with this recommendation and we removed this sentence.

18 Table 1 In variables such as age, BMI and several others usually more informative for descriptive purposes to give the minimum and maximum values rather than quoting the SD.

For gender and many variables below the numbers are integers so the decimal points should be omitted. Also for variables using the median, better to use the minimum and maximum values rather than the IQR.

We thank the reviewer for these observations. We modified the Table 1 according to his recommendations.

** Table 2 I suspect the unadjusted differences in the final column of this table would be very similar to the adjusted values.Adjusted values might be helpful if they substantially change the unadjusted values and hence substantially change the interpretation of the findings. I suspect « substantial changes » will not be the case here. I recommend, the authors to consider quoting the unadjusted values in table 2 (and in Appendix B) If these changes are made then Page 6, Statistical Methods, lines 2-5 and 8-9 can be simplified.

We refer the reviewer to our comment above (we followed EMA recomandations). An alternative could be to perform both unadjusted and adjusted analysis but we don’t think that this would make the reading easier.

Reviewer #2: Excellent idea, it trigger a new perspective, the research was conducted in an efficient way, results are clear enough, discussion section is written in a very good manner, conclusion is also clearly mentioned. Well done.

Reviewer #3: The manuscript is well done. Here are some concerns that should be addressed.The significant drop-off in participant numbers from eligibility to analysis (1309 eligible vs. 536 analyzed) raises concerns about participant engagement and retention. Discussing the reasons for attrition and potential biases introduced by this attrition is crucial for interpreting the study's findings accurately.

We thank the reviewer for this recommendation. We deleted the reference to drop-off in the main results discussion to develop this bias in the limit section and added qualitative material from ancillary qualitative studies. We rephrased the paragraph as followed : « Attrition bias is a major issue in this study due to a lower than the expected number of enrolments, a large number of unexpected dropouts early in the study. Despite the intervention being free of charge implemented out of working hours, and close to participant living area, 59% of eligible participants chose not to participate. When writing a research protocol involving behavior change, it is commonly accepted that an attrition of 20% should be anticipated. However, attrition rates in motivational studies are extremely variable. Many studies do not provide their attrition rate. A Chinese study aimed at improving blood pressure over 6 months reported an attrition rate of 0.08% (1). At the extreme opposite, a Scandinavian 12-month study aimed at improving professional activity in the workplace had an attrition rate of 78% (2). In the SPICES study, non-participants reported competing priorities of different kinds. They could be family caregivers, have new professional constraints, or have a non-cardiovascular health event. In qualitative interviews, some dropouts were related to the time-consuming nature of the intervention, the nature of group intervention, a feeling of déjà vu of cardiovascular prevention messages, the relationship with champions (3,4). Maybe participants with moderate cardiovascular disease may be less motivated to make behavioural changes. Most previous studies on behavioural change selected participants with high cardiovascular risk who are most likely to notice an immediate, measurable health benefit. »

The specific impacts of the COVID-19 pandemic on the study, such as disruptions to intervention delivery, participant follow-up, and data collection, should be elaborated upon. How these challenges were managed, and their potential influence on study outcomes need to be thoroughly discussed.

The impacts of the Covid-19 pandemic on the survey are presented in the Conserve-Consort extension section. A new paragraph has been added to the limits section to address the reviewer’s recommendation. We hope it will meet his expectations : 

« Confusion biases:

In France, the Covid- 19 related health measures included three strict lockdowns and gradual deconfinements, restricting public meetings for months. Group meetings were interrupted at each lockdown and resumptions were slowed by local stakeholders. The qualitative study of dropouts highlighted the difficulty of maintaining physical activity due to lockdowns, an upheaval of personal priorities induced by the pandemic and a saturation of prevention messages induced by the omnipresence of messages to limit the transmission of Covid (3,4). Citizens and participants did not wish to dematerialize the meetings. Data collection was dematerialized, allowing collection to be ensured according to the protocol. »

Despite community champions leading behavioral change sessions, the primary outcome (INL score) did not show a significant difference between intervention and control groups at 24 months. Potential reasons for this lack of significant effect, including the nature of the behavioral goals chosen by participants and the intensity or duration of the intervention, must be discussed.

We thank the reviewer for this comment however these specific points were already addressed in the main results discussion section : « Apart from issues involved with implementing the intervention, the appropriateness of using the INL as a primary measurement tool for cardiovascular risk is questionable. INL was chosen for its ease of use in a community setting, its having heavily weighted items like smoking and lighter weighted items like diet that participants could easily improve. Nevertheless, the INL did not capture small improvements in stress and depression. Similarly, another preventive study found that the waist-to-hip ratio, included in the INL to evaluate obesity as a cardiovascular risk, did not change (5). 

Interestingly, most participants in the SPICES Intervention group set themselves a goal to lose weight. Yet the mean BMI was 27.06± 5.01, which is slightly higher than the mean BMI for women (24.2) and men (25.5) in France in 2014 (6). Notably, weight-loss for people with a BMI of around 27 is not recommended, because they risk having weight rebound and greater weight gain (7). Moreover, published studies on weight-loss strategies often show very small benefits with no change in obesity/overweight class and involve small populations (5), (8). In comparison, 16% of the cohort were current smokers, and only 25% of Intervention group smokers tried to stop smoking. Yet quitting smoking is a change which provides major health benefits whereas health benefits from losing weight are lower. This learning from the SPICES project is crucial when promoting health amongst the general population. There is a need to support people in their health choices to redirect their decisions towards effective strategies. »

Clarify the validity and reliability of the measurement tools used to assess outcomes, such as the WHOQOL-BREF for quality of life, the DASH questionnaire for diet, and the IPAQ-short for physical activity. Ensuring these tools are appropriate and validated for the study population strengthens the reliability of reported findings

These points were adressed in a previous publication. The Spices protocol has been published in 2021. The following sentence has been added to the current publication (9) : « The validity and reliability of the measurement tools have been presented in a previous publication. »

The reviewer will find below the content of the protocol detailing the measurement tools : « The secondary outcomes after 24 months will be quality of life, as assessed by the WHOQOL-BREF (17), modification of diet, according to the DASH questionnaire (18), modification of physical activity, following the IPAQ-short (19), BMI reduction, smoking level, modification of self-declared alcohol consumption. 

• The WHOQOL-BREF questionnaire was derived from the WHOQOL-100 in 1996 to assess quality of life by using thirty-three questions within the four following domains: physical health, psychological domain, social relationships and environment. 

• The DASH questionnaire was created in 2016 to assess, by means of eleven questions, the quality of an individual’s diet, according to the Dietary Approach to Stop Hypertension. This diet was designed along the lines of the Mediterranean diet to improve the cardiovascular risk profile of the consumers (20). 

• The IPAQ was created in 2003 to assess physical activity. The short version comprises seven questions in four blocks which describe intense activity, moderate activity, walking in the last seven days and time spent seated. The score of the IPAQ-short can be converted in Metabolic Equivalents of Task, a standardised measurement of physical activity level. »

---

## [Decision Letter · Decision Letter 1]

18 Nov 2024

Innovative population-based strategies for primary prevention of cardiovascular disease: A 2-year randomised control trial evaluating behavioral change led by community champions versus brief advice.

PONE-D-23-42871R1

Dear Dr. Le Goff,

We’re pleased to inform you that your manuscript has been judged scientifically suitable for publication and will be formally accepted for publication once it meets all outstanding technical requirements.

Kind regards,

Mohd Akbar Bhat

Academic Editor

PLOS ONE

Additional Editor Comments (optional):

Reviewers' comments:

Reviewer's Responses to Questions

**Comments to the Author**

1. If the authors have adequately addressed your comments raised in a previous round of review and you feel that this manuscript is now acceptable for publication, you may indicate that here to bypass the “Comments to the Author” section, enter your conflict of interest statement in the “Confidential to Editor” section, and submit your "Accept" recommendation.

Reviewer #1: All comments have been addressed

Reviewer #3: All comments have been addressed

2. Is the manuscript technically sound, and do the data support the conclusions?

Reviewer #1: (No Response)

Reviewer #3: No

3. Has the statistical analysis been performed appropriately and rigorously? 

Reviewer #1: (No Response)

Reviewer #3: No

4. Have the authors made all data underlying the findings in their manuscript fully available?

Reviewer #1: (No Response)

Reviewer #3: Yes

5. Is the manuscript presented in an intelligible fashion and written in standard English?

Reviewer #1: (No Response)

Reviewer #3: Yes

6. Review Comments to the Author

Reviewer #1: (No Response)

Reviewer #3: After reviewing the manuscript, I found it does not provide sufficient evidence to support the efficacy of the intervention being tested. Here are several critical issues:

The primary outcome, which measures the change in the Non-laboratory Interheart risk score (INL), shows no significant difference between the intervention and control groups at 24 months. This suggests that the intervention was not effective in improving cardiovascular health compared to brief advice alone.

There is a significant drop in participant numbers from enrollment to analysis (from 1309 eligible participants to 536 analyzed). This high dropout rate might have introduced substantial attrition bias, which is not adequately addressed or compensated for in the analysis.

Most participants chose weight loss as their health goal, which was not an effective goal for the population with a mean BMI at the edge of overweight and not obese. This misalignment between participant goals and effective cardiovascular risk reduction strategies may have further diluted any potential impact of the intervention.

7. PLOS authors have the option to publish the peer review history of their article (what does this mean?). If published, this will include your full peer review and any attached files.

Reviewer #1: No

Reviewer #3: No

---

## [Editor Report · Acceptance letter]

2 Dec 2024

PONE-D-23-42871R1 

PLOS ONE

Dear Dr. Le Goff, 

I'm pleased to inform you that your manuscript has been deemed suitable for publication in PLOS ONE. Congratulations! Your manuscript is now being handed over to our production team.

Kind regards, 

on behalf of

Dr. Mohd Akbar Bhat 

Academic Editor

PLOS ONE